# Pancreatic Ductal Adenocarcinoma: Molecular Pathology and Predictive Biomarkers

**DOI:** 10.3390/cells11193068

**Published:** 2022-09-29

**Authors:** Mehran Taherian, Hua Wang, Huamin Wang

**Affiliations:** 1Department of Anatomical Pathology, The University of Texas MD Anderson Cancer Center, Houston, TX 77030, USA; 2Department of Gastrointestinal Medical Oncology, The University of Texas MD Anderson Cancer Center, Houston, TX 77030, USA; 3Department of Translational Molecular Pathology, The University of Texas MD Anderson Cancer Center, Houston, TX 77030, USA

**Keywords:** pancreatic ductal adenocarcinoma, molecular pathology, predictive marker, tumor microenvironment, targeted therapy and immunotherapy

## Abstract

Pancreatic ductal adenocarcinoma (PDAC) has an extremely poor prognosis due to the lack of methods or biomarkers for early diagnosis and its resistance to conventional treatment modalities, targeted therapies, and immunotherapies. PDACs are a heterogenous group of malignant epithelial neoplasms with various histomorphological patterns and complex, heterogenous genetic/molecular landscapes. The newly proposed molecular classifications of PDAC based on extensive genomic, transcriptomic, proteomic and epigenetic data have provided significant insights into the molecular heterogeneity and aggressive biology of this deadly disease. Recent studies characterizing the tumor microenvironment (TME) have shed light on the dynamic interplays between the tumor cells and the immunosuppressive TME of PDAC, which is essential to disease progression, as well as its resistance to chemotherapy, newly developed targeted therapy and immunotherapy. There is a critical need for the development of predictive markers that can be clinically utilized to select effective personalized therapies for PDAC patients. In this review, we provide an overview of the histological and molecular heterogeneity and subtypes of PDAC, as well as its precursor lesions, immunosuppressive TME, and currently available predictive molecular markers for patients.

## 1. Introduction

The annual incidence of pancreatic cancer has increased worldwide, with 495,773 new cases reported in 2020 [1]. Pancreatic cancer is projected to be the second leading cause of cancer-related death for both men and women in the United States by 2030 [2]. Pancreatic ductal adenocarcinoma (PDAC), the most common type of pancreatic cancer, has the most dismal prognosis among all solid tumors, with a 5-year survival rate of approximately 10% [3,4]. Due to its aggressive biology, lack of distinctive clinical symptoms, and lack of reliable biomarkers for early detection and diagnosis, most patients with PDAC present with locally advanced or metastatic disease, which is not suitable for a potentially curative surgical resection [5]. In addition, PDAC is resistant to conventional treatment modalities, including chemotherapy and radiotherapy, and it has limited response to the modern targeted therapies and immunotherapies; therefore, the prognosis and survival for patients with PDAC have not significantly changed over the years [6]. The histological subtypes and precursors of PDACs are associated with distinctive genetic alterations, histomorphological features, and different prognoses [7]. The lack of effective treatment and grim prognosis results from our poor understanding of the complicated and multifactorial nature of PDAC biology, the immunosuppressive tumor microenvironment (TME), and the molecular genetic heterogeneity among primary tumor cells and among metastasis-initiating cells [8]. In recent years, significant progress has been made in profiling the molecular alterations and classification of PDAC, but these findings are yet to be translated into early diagnosis and effective therapy. Therefore, a better understanding of the histological, genetic and molecular heterogeneities; the aggressive biology and the immunosuppressive TME; and the development of predictive and prognostic markers are important in the development of effective personalized therapies. In this review, we provide an overview of the histological and molecular subtypes of PDAC and its precursor lesions, its immunosuppressive TME, and the predictive molecular markers for PDAC treatment.

## 2. Histology and Morphologic Heterogeneity of PDAC

PDACs are a heterogeneous group of malignant pancreatic epithelial neoplasms. Conventional PDACs are characterized by dense desmoplastic stroma intermixed with angulated glands, small nests of malignant epithelial cells and/or single tumor cells (Figure 1A). PDACs often show a spectrum of differentiation ranging from well-differentiated to poorly differentiated adenocarcinomas within the same tumor and show significant inter- and intra-tumoral heterogeneity in histomorphological patterns, such as complex interconnecting tumoral glands embedded in desmoplastic stroma (Figure 1B), a large duct type (Figure 1C), poorly differentiated carcinoma with eosinophilic or clear cells (Figure 1D–F), complex intraluminal micropapillae formation (Figure 1G), cribriform histology with foamy cells (Figure 1H), and the pagetoid involvement of pancreatic duct/intraductal carcinoma (Figure 1I). Aggressive histological features, such as lymphovascular invasion, tumor invasion into peripancreatic soft tissue, large vessels and adjacent organ(s)/structure(s), perineural invasion, the involvement of resection margins, and lymph node metastasis, are frequently present in resected PDAC specimens (Figure 2A–F). The presence of these aggressive histological features is associated with an increased risk of post-operative tumor recurrence/distant metastasis and poor survival in PDAC patients [9,10,11,12]. 

In addition to conventional PDAC, there are nine histological subtypes of PDAC according to the World Health Organization (WHO) classification, which further highlight the morphologic heterogeneity of PDAC [13]. The subtypes of PDAC are adenosquamous carcinoma (ASC)/squamous cell carcinoma (SCC), colloid carcinoma, hepatoid carcinoma, signet ring cell (poorly cohesive cell) carcinoma, undifferentiated carcinoma, undifferentiated carcinoma with osteoclast-like giant cells, medullary carcinoma, micropapillary carcinoma, and undifferentiated carcinoma with rhabdoid cells (rhabdoid carcinoma) (Figure 3A–I). While some PDAC subtypes share a similar molecular pathogenesis, biological and clinical behavior, and prognosis to conventional PDAC, these subtypes are characterized by specific histomorphological and clinical features, and some have a different molecular profile, genetic alterations, and prognosis. For example, colloid carcinoma typically arises in association with an intestinal type of intraductal papillary mucinous neoplasm (IPMN) and has a better prognosis than conventional PDAC [14,15,16,17]. On the other hand, ASC and undifferentiated carcinomas have worse prognoses than conventional PDAC [13,15,18]. Medullary carcinoma is more frequently associated with a high level of microsatellite instability, which may predict better responses to immunotherapy [19,20], and often with the wild-type *KRAS* gene [21,22,23]. The subtypes of PDAC and their associated specific molecular/genetic alterations are listed in Table 1. 

## 3. Genetic Alterations and Molecular Subtypes of PDAC

PDAC is characterized by a handful of inherited (germline) and recurring somatic mutations. The first whole exome sequencing of human PDAC samples was reported in 2008 by Jones et al. [24]. In that study, 20,661 protein-coding genes in 24 PDAC samples were sequenced, and more than 1500 somatic mutations in 1007 genes were identified. This study was followed by several landmark, large-scale whole exome sequencing and comprehensive molecular profiling of human PDAC samples, which provided us with the in-depth understanding of the heterogeneous molecular landscapes of PDAC [25,26,27]. These studies identified four “mountains” (the genes mutated at the greatest frequency): oncogenic mutations of *Kirsten rat sarcoma (KRAS*), loss-of-function mutations and/or deletions of the *TP53* tumor suppressor genes, *mothers against decapentaplegic homolog 4 (SMAD4*), and *the cyclin dependent kinase inhibitor 2A (CDKN2A*) [24]. The data from genetically engineered mouse models have shown that these mutations play an essential role in the development and/or progression of PDAC [24,28,29]. In addition to the four “mountains”, a large number of less common “hills” (genes mutated at low frequencies) have been detected in PDACs. For example, amplifications of other less frequent oncogenes such as *CMYC* (on chromosome 8q), *MYB* (chromosome 6q), *AIB1/NCOA3* (chromosome 20q), *EGFR* (chromosome 7p), and *GATA6*, as well as recurrent chromosomal amplifications, have also been identified [7]. One or more somatic/germline mutations of the genes involved in DNA damage repair (DDR), such as *BRCA2, BRCA1*, *PALB2*, *ATM*, *CHEK2*, *RAD51C,* and *RAD51D* mutations, may be detected in 10–20% of PDAC patients [30,31,32,33]. These less common genetic alterations may represent valuable targets or serve as predictive biomarkers for PDAC patients. For example, defects in DDR pathway in PDACs represent a unique subset of patients who may benefit from platinum-based chemotherapy (e.g., cisplatin) or the newly approved poly (ADP-ribose) polymerase (PARP) inhibitors such as olaparib [34,35]. 

Multiple studies have reported on the molecular subtypes of PDAC based on the whole exome sequencing data and/or integrated analyses of the genomic, transcriptomic, proteomic, and epigenetic profiles of human PDAC samples [36,37,38,39]. Collisson et al. reported three molecular subtypes of PDAC: the classical, quasi-mesenchymal (QM), and exocrine-like based on the analysis of 27 microdissected PDAC samples [38]. The classical subtype showed a higher expression of adhesion-associated and epithelial genes and a higher expression of *KRAS* and *GATA6,* an essential gene for pancreatic development and PDAC progression, compared with the QM subtype. The QM subtype was found to have a high expression of mesenchyme-associated genes. The exocrine-like subtype showed the relatively high expression of tumor cell-derived digestive enzyme genes. These molecular subtypes were found to be significantly correlated with patient survival in that the QM subtype had the worst survival. They also demonstrated that PDAC cell lines of the QM subtype were more sensitive to gemcitabine, whereas the classical subtype cell lines were more sensitive to erlotinib (an *EGFR* antagonist) [38]. Using non-negative matrix factorization to digitally dissect the tumor and stromal gene signature of primary and metastatic PDAC, Moffitt et al. identified two tumor-specific subtypes: the basal-like subtype, which is molecularly similar to the basal-like carcinoma of breast and bladder, and the classical subtype, and two stromal subtypes: normal and activated. The basal-like subtype had a worse prognosis but a superior response to adjuvant therapy compared with the classical subtype [39]. Via RNA sequencing, they showed that the *KRAS*^G12D^ mutation was significantly overrepresented in the basal-like subtype, *KRAS*^G12V^ was isolated to the classical subtype, and *SMAD4* expression was significantly higher in the classical subtype compared with the basal-like subtype, which is consistent with the observation that SMAD4 loss confers a more aggressive tumor behavior [39]. The normal stroma showed the relatively high expression of known markers for pancreatic stellate cells (desmin, smooth muscle actin, and vimentin), whereas the activated stroma was characterized by a gene set associated with macrophages (integrin *ITGAM* and the chemokine ligands *CCL13* and *CCL18*) and other genes that are reported to promote tumor progression (*SPARC*, *WNT2*, *WNT5A*, *MMP9*, and *MMP11*) [39,40]. Patients with the classical subtype and activated stroma had worse survival compared with those with the classical subtype and normal stroma. Stromal subtypes were not associated with survival in patients with basal-like subtype PDAC [39]. Moffitt’s classification of PDACs into basal-like and classical subtypes was validated by Puleo et al., who classified PDACs into five subtypes: pure basal-like, stroma-activated, desmoplastic, pure classical, and immune classical based on features of cancer cells and the tumor microenvironment [41]. Thus, targeting the tumor-promoting genes in activated stroma may represent a potential strategy for PDAC patients.

The integrated genomic analysis of 456 PDAC samples by Bailey et al. identified 32 recurrent mutated genes in 10 pathways—*KRAS*, *TGF-β*, *WNT*, *NOTCH*, *ROBO/SLIT* signaling, *G1/S* transition, *SWI-SNF*, chromatin modification, DNA repair, and RNA processing—and four molecular subtypes of PDACs—pancreatic progenitor, squamous, aberrantly differentiated endocrine exocrine (ADEX), and immunogenic [36]. The pancreatic progenitor subtype (19%) was characterized by the transcriptional factors involved in early pancreatic development (*PDX1*, *MNX1*, *HNF1A*, *HNF1B*, *HNF4A*, *HNF4G*, *FOXA2*, *FOXA3*, and *HES1*) and metabolic pathways such as fatty acid oxidation and drug metabolism [42]. The squamous subtype (31%) was enriched with *TP53* and *KDM6A* mutations, as well as the hypermethylation of genes governing pancreatic endodermal cell-fate determination (e.g., *PDX1*, *MNX1*, *GATA6*, and *HNF1B*). The ADEX subtype (21%) showed the upregulation of genes that regulate networks involved in *KRAS* activation. This subtype included two gene programs, with one focused on exocrine function (*NR5A2*, *MIST1* and *RBPJL*) and the other related to endocrine differentiation (*NEUROD1*, *MODY*, *INS* and *NKX*2–2). The immunogenic subtype (29%) contained a family of genes related to immune cell function including B cell signaling, Toll-like receptor signaling, antigen presentation, and the infiltration of CD8+ and CD4+ T cells, with the upregulation of the immune inhibitor PD-1 and cytotoxic T-lymphocyte-associated protein-4 (CTLA-4) [36,42]. These molecular subtypes correlated with histopathological features in that the squamous subtype represented ASC, progenitor and immunogenic represented colloid carcinomas and carcinomas arising from IPMN, and ADEX represented rare acinar cell carcinomas [36]. Among these molecular subtypes, the squamous subtype had the worst survival, while the other three subtypes showed similar survival rates [36]. Another integrated analysis of the mRNA, miRNA, lncRNA and DNA methylation profiling of 150 PDAC samples by the Cancer Genome Atlas Research Network identified two subtypes of PDACs: SNF-1 and SNF-2. The SNF-1 subtype represented most of the basal-like subtype in Moffitt’s classification, the squamous subtype in Bailey’s classification, and the QM subtype in Collison’s classification [37]. 

More recently, Chan-Seng-Yue et al. performed whole genome and transcriptome analysis of purified tumor cells from 314 primary and metastatic PDAC patients to generate tumor-specific expression signatures [43]. They classified PDACs into five molecular subtypes: basal-like A and B for the previously defined basal-like subtype, hybrid, and classical A and B for the previously defined previously defined classical subtype. The hybrid subtype was inconsistently classified by previous classification systems due to the presence of multiple expression signatures. Patients with basal-like A PDAC often present with advanced disease and show the worst response to gemcitabine-based chemotherapies and FOLFIRINOX. In contrast, patients with basal-like B and hybrid tumors often present with resectable disease. Therefore, the ability to distinguish the basal-like A, basal-like B, and hybrid subtypes from the group formerly classified as basal-like allows for the more accurate prediction of chemotherapy response. Classical A/B tumors were found to be associated with an increased frequency of *GATA6* amplification and complete SMAD4 loss, whereas basal-like A/B tumors showed the complete loss of *CDKN2A* and a higher frequency of *TP53* mutations. At single-cell resolution, the authors also showed that basal-like and classical subtypes can co-exist in the same tumor, which highlighted the intra-tumoral molecular heterogeneity [43]. 

The most recent molecular subclassification of PDACs was reported by Hwang et al. in 2022. Using the single-nucleus RNA sequencing and whole digital spatial transcriptome profiling of 43 primary PDAC samples (18 untreated and 25 treated), they identified three distinct subtypes: classical, squamoid–basaloid, and treatment-enriched. Their study uncovered that the neural-like progenitor (NRP) malignant cell program was enriched in residual carcinoma after chemoradiation therapy. The NRP cells were associated with treatment resistance and poor survival in PDAC patients via the regulation of genes involved in drug efflux, the negative regulation of cell death, and resistance to chemotherapy (e.g., *ABCB1*, *BCL2*, *PDGFD* and *SPP1*), tumor–nerve crosstalk (e.g., *SEMA3E*, *RELN* and *SEMA5A*), and metastasis (*NFIB*) [44].

These molecular classifications of PDAC provide rich and comprehensive datasets to better understand pancreatic tumorigenesis, genetic/molecular landscapes, intra- and inter-tumoral heterogeneity, tumor progression, and drug resistance. More importantly, the molecular subtyping of PDAC may provide useful information for more effective subtype-tailored therapies for PDAC patients. However, due to their complexity, these classifications of PDAC have not been utilized in daily pathologic diagnosis or clinical practice.

## 4. Heterogeneous Response of PDAC to Neoadjuvant Therapy

Neoadjuvant therapy is routinely used to treat PDAC patients with borderline resectable and high-risk resectable disease, as well as selected patients with locally advanced disease [45]. Pathologic studies of the post-therapy pancreatectomy specimens have shown that only 12.6% to 18.6% PDAC patients demonstrate a complete or near complete pathologic response to neoadjuvant therapy, which is associated with better survival, while the majority of PDAC patients (>80%) demonstrate a moderate or minimal response to neoadjuvant therapy and poor survival [46,47,48,49]. These data highlight not only the fact that vast majority of PDACs are resistant to neoadjuvant chemotherapy with or without radiation but also the inter-tumoral heterogeneity in tumor response among PDAC patients. It is also not uncommon to observe significant intra-tumoral, heterogeneous response to neoadjuvant therapy in different areas of the same treated tumor, with some areas of the tumor showing complete or near complete response and other areas showing minimal or no response (Figure 4A,B). Occasionally, differential responses to neoadjuvant therapy are also observed between the primary PDAC and the metastatic carcinoma in lymph node(s) in the same patient (Figure 4C,D). Currently, there are limited data on the molecular correlation with tumor response to neoadjuvant therapies. The molecular mechanisms underlying the inter- and intra-tumoral heterogeneity in response to neoadjuvant therapy is not clear. It is possible that the cellular and molecular/genetic heterogeneity and the heterogeneity in the TME contribute to the heterogeneous response in PDAC patients. The recent molecular profiling of treated PDACs suggested that the NRP malignant program was enriched in residual carcinoma after chemoradiation therapy and plays a role in tumor resistance to therapy [44]. Future biomarker-driven clinical trials of neoadjuvant therapy are needed to address this important question.

## 5. Precursor Lesions of PDAC

According to the WHO tumor classification, the histologically recognized precursor lesions of PDAC include pancreatic intraepithelial neoplasia (PanIN), IPMN, intraductal oncocytic papillary neoplasms (IOPN), intraductal tubulopapillary neoplasm (ITPN), and mucinous cystic neoplasm (MCN) [13]. These lesions have different clinical and pathologic features, and they are often associated with different grades of dysplasia and molecular characteristics, which are summarized in Figure 5.

PanIN lesions are microscopic papillary or flat noninvasive epithelial neoplasms (<0.5 cm) arising in pancreatic ducts composed of mucin-containing cuboidal-to-columnar cells [50,51]. PanINs are further graded based on the highest degree of cytologic and architectural atypia as low grade or high grade [13]. Molecular analyses have demonstrated that PanIN lesions share critical genetic abnormalities with adjacent PDAC, e.g., >90% of PanIN lesions of all grades harbor *KRAS* mutations [52,53,54,55]. Telomere shortening and *KRAS* oncogene mutations are early genetic events more observed in low-grade PanIN, whereas the biallelic inactivation of *CDKN2A/P16*, the loss of SMAD4, and p53 mutations are found in high-grade PanINs [53,56].

IPMNs are mucin-producing epithelial neoplasms, usually with a papillary architecture. Three distinct subtypes have been identified by imaging: main duct-type, branch duct-type, and mixed duct-type IPMN [57]. These lesions are larger than PanINs (≥1 cm), and they are classified into low and high grade. Low-grade IPMNs show mild-to-moderate atypia with or without papillary projections and mitoses. High-grade IPMNs are composed of cells with marked nuclear atypia and a loss of polarity, papillae with irregular branching and budding, and frequent mitosis. Based on the histological and immunohistochemical features, IPMNs can be subclassified into gastric (~70% of the cases), intestinal (~20%), and pancreatobiliary subtypes [58,59,60,61]. Most gastric-type IPMNs are low grade and associated with the lowest risk of invasion in contrast to the intestinal and pancreatobiliary types that often display high-grade dysplasia [13]. *GNAS* (50–70% of IPMNs), *KRAS* (60–80%), and *RNF43* (~50%) are the most common mutations found in IPMN. Mutations in other driver genes, including *CDKN2A*, *TP53*, and *SMAD4,* more frequently occur in IPMNs with high-grade dysplasia or associated invasive carcinoma [62]. The overexpression of p53, a surrogate for missense mutations of *TP53*, can be found in 10–40% of high-grade IPMNs and 40–60% of invasive carcinomas associated with IPMN [63,64,65,66]. The loss of SMAD4 typically occurs in the context of invasion [67,68]. Ductal lesions between 0.5 cm and 1.0 cm could represent either dilated ducts lined by PanIN or small IPMNs. The term “incipient IPMN” was proposed for lesions with long finger-like papillae, villous intestinal or oncocytic differentiation, or a GNAS mutation [69].

In IOPN, the papillae are lined by multiple layers of cuboidal-to-columnar cells with an eosinophilic granular cytoplasm, a round nucleus, and a prominent nucleolus. IOPNs often show cribriform structures and intraluminal mucin formation [13,70]. IOPNs, unlike IPMN, typically lack mutations in *KRAS*, *GNAS*, and *RNF43* [71]. Genes including *ARHGAP26*, *ASXL1*, *EPHA8* and *ERBB4* are somatically mutated in some IOPNs [71]. A recent study of 20 IOPNs identified fusions in *PRKACA* and *PRKACB* genes similar to those identified in the fibrolamellar variant of hepatocellular carcinoma [72].

ITPNs histologically show circumscribed nodules of back-to-back tubules surrounded by fibrotic stroma. The tubules are lined by cuboidal or low columnar epithelium, with a modest amount of eosinophilic or amphophilic cytoplasm, but lack mucin production. ITPNs are architecturally complex and have high-grade dysplasia [13]. In general, ITPNs lack *KRAS*, *GNAS* and *BRAF* mutations [73]. However, mutations have been described in chromatin-remodeling genes (*MLL1*, *MLL2*, *MLL3*, *BAP1*, *PBRM1*, *EED*, and *ATRX*), the PI3K pathway (*PIK3CA*, *PIK3CB*, *INPP4A*, and *PTEN*), and a minority of *FGFR2* and *STRN–ALK* fusions [73].

MCNs predominantly occur (>98%) in women [74]. In contrast to IPMNs, MCNs do not have a connection with the pancreatic duct. In addition, MCNs are unique among pancreatic precursor lesions because the cysts have an underlying ovarian-type stroma [13]. The epithelial component of MCNs harbors activating mutations in codon 12 of *KRAS* in 50–66% of cases, as well as loss-of-function alterations in *RNF43* [75,76,77]. Mutations of *TP53* are rare and may be seen in more advanced MCNs [75,76].

Simple mucinous cysts are pancreatic cysts > 1.0 cm lined by a gastric-type flat mucinous epithelium with minimal atypia without ovarian-type stroma. In rare instances, focal high-grade dysplasia may be present. KRAS mutations can be detected in these cysts, suggesting the possibility that these lesions could represent another precursor of PDAC [78,79].

## 6. Emerging Predictive Markers and Targeted Therapies for PDAC Patients

Currently, the main treatment options for patients with metastatic PDAC are gemcitabine/nab-paclitaxel, modified folinic acid, fluoracil, irinotecan, oxaliplatin (FOLFIRINOX), gemcitabine/capecitabine, irinotecan/fluorouracil, and single gemcitabine [80,81]. These chemotherapy regimens show modest efficacy. Compared to non-small cell lung cancer (NSCLS), colorectal cancer (CRC), and breast cancer, there are very limited number of predictive markers that are currently in clinical use for PDAC patients. With advances in the germline testing and molecular profiling of PDACs, few new predictive markers are emerging and helping oncologists to select the best possible personalized treatment for PDAC patients.

### 6.1. Markers for the Defective DNA Damage Responses

A significant proportion of PDACs (~10%) harbor either somatic or germline mutations in DNA damage response (DDR) genes, such as *BRCA1*, *BRCA2, PALB2* and *ATM*. Overall, *ATM* appears to be the most frequently mutated DDR gene in somatically mutated sporadic PDAC, followed by *BRCA2*, *STK11* and *BRCA1* [82]. PDACs with defective DDR may be vulnerable to new therapeutic agents, such as platinum and poly (ADP-ribose) polymerase (PARP) inhibitors, which may cause DNA damage at a level beyond the tolerable threshold and cell death. Patient-derived PDAC cell lines with deficient DDR were found to be more sensitive to both cisplatin therapy (*p* = 0.031) and PARP inhibition (*p* < 0.001) compared with those with proficient DDR [83]. Golan et al. demonstrated that patients with metastatic PDAC and *BRCA* germline mutations who received first-line platinum-based chemotherapy followed by the maintenance therapy of olaparib (a potent PARP inhibitor) had significantly better progression-free survival rates [84]. Other DDR deficiencies such as *ATM* inactivation have also been shown to significantly improve sensitivity to chemotherapy and PARP inhibitors [85,86]. Two selective *ATM* inhibitors, AZD0156 and AZD1390, were shown to increase cell cycle arrest and apoptosis in preclinical studies [87,88]. Of note, PDACs with deficient mismatch repair (MMR) or MSI-high showed promising response to immune checkpoint inhibitors [89]. Therefore, the new biomarker-driven National Comprehensive Cancer Network (NCCN) guidelines suggest that the PARP inhibitor, olaparib, may be used as one of the maintenance therapies in patients who have a germline BRCA1 or BRCA2 mutation and received first-line platinum-based chemotherapy without disease progression for at least 16 weeks.

### 6.2. KRAS^G12C^ Mutation

The *KRAS*^G12C^ mutation was reported in 1.3% of PDACs, 1–3% of CRCs, and 13% of NSCLCs [90]. A recent phase I trial of a selectively small molecular inhibitor of *KRAS*^G12C^ mutation, sotorasib, showed a promising response in patients with advanced NSCLC, CRC, PDAC, and carcinoma from the appendix and endometrium, as well as melanoma [91]. The recent CodeBreaK100 phase I/II single arm trial for sotorasib of 38 patients with stage IV PDAC who had received at least one therapy showed an objective response rate of 21.1% and a disease control rate of 84.2% [92]. These data suggested that the selective *KRAS*^G12C^ inhibitor sotorasib could be used to treat eligible patients with metastatic PDAC and *KRAS*^G12C^mutation. It should be noted that the *KRAS*^G12C^ mutation is only present in a very small percentage of PDAC patients, even though *KRAS* is the most frequently mutated gene reported in >90% PDACs. Recent studies showed that selective inhibitors targeting *KRAS^G12D^*, MRTX1133, and BI-KRASG12D1–3 can interact with *KRAS^G12D^* and inhibit the proliferation and viability of tumors that harbor *KRAS*^G12D^ but not the tumor cells with wild-type *KRAS* in both in vitro studies and in vivo preclinical xenograft models [90]. Exciting progress has already been made in the development of both selective inhibitors targeting *KRAS* mutations other than *KRAS^G12C^* and pan-*KRAS* inhibitors and degraders that target a broad range of KRAS alterations, including *KRAS*^G12D^, *KRAS*^G12V^, *KRAS*^G13D^, *KRAS*^G12R^, *KRAS*^G12A^, and the amplification of wild-type *KRAS*. If successful, these *KRAS* mutation-selective inhibitors and pan-*KRAS* inhibitors and degraders will move beyond selective inhibitors of *KRAS^G12C^* and provide novel therapeutics for not only PDAC patients but also all other patients with *KRAS*-driven cancers such as NSCLCs and CRCs.

### 6.3. Neurotrophic Tropomyosin Receptor Kinase (NTRK) Fusions

Targeting the oncogenic driver(s) using Herceptin, tyrosine kinase inhibitors (TKIs), or MEK inhibitors has been shown to be effective in treating patients with breast cancer, NSCLC, CRC, or melanoma. These targeted therapies, however, have either no or limited response in PDAC patients. The fusions of the NTRK carboxy-terminal tyrosine kinase domain with different genes through either intrachromosomal or interchromosomal rearrangements have been identified in a small percentage of PDACs (<1%) and other types of malignancies [93]. The chimeric protein from NTRK fusions is ligand-independent and constitutively active, and it plays an important role in cell proliferation and oncogene addiction [93]. NTRK fusions can be detected in tumor samples by fluorescence in situ hybridization (FISH), next-generation sequencing, or immunohistochemistry, and they can used as predictive markers for selecting PDAC patients to receive selective inhibitors for NTRK. *NTRK* fusion-positive solid tumors showed durable and clinically meaningful responses to selective inhibitors of NTRK, larotrectinib and entrectinib, in clinical trials [93,94]. These results highlight the importance of the routine screening for NTRK fusion in PDAC samples to identify patients who may benefit from selective inhibitors targeting NTRK fusion.

### 6.4. Microsatellite Instability (MSI)/Defective Mismatch Repair (dMMR)

The MSI/dMMR is present in a small percentage (1–2%) of PDACs and is strongly associated with wild-type *P53* and *KRAS* and with medullary carcinoma or colloid carcinoma of the pancreas. MSI/dMMR in PDAC patients may occur either in association with Lynch syndrome/hereditary non-polyposis colorectal cancer syndrome (HNPCC) or as a sporadic type, which is often due to the hypermethylation of the promoter region of the MLH1 gene. Multiple recent phase II clinical trials demonstrated that MSI/dMMR is a strong predictive marker for tumor response to immunotherapy in patients with carcinomas of different origins [19,95]. In 2017, the Food and Drug Administration (FDA) approved the use of pembrolizumab to treat patients with MSI-high tumors solely based on the MSI status, not the primary origin. Pembrolizumab and/or nivolumab therapies induce durable responses and long progression-free and overall survival in patients with metastatic or recurrent MSI/dMMR CRCs [96,97]. The responses of MSI/dMMR tumor are due in part to the presence of high tumor mutational burden (TMB) (which gives rise to significantly higher levels of tumor neoantigens than microsatellite stable (MSS) tumors) and to antitumor immune responses such as the increased infiltration and activation of cytotoxic T cells and T_H_1 cells with interferon-γ (IFN γ) production. Although large clinical trials of immunotherapy for patients with MSI/dMMR PDAC have not been reported due to the rarity of MSI/dMMR PDACs, a robust 62% response rate to pembrolizumab was observed in patients with MSI/dMMR PDAC in recent clinical trials [19,95]. The molecular characterization of MSI/dMMR PDAC showed that these tumors have a high prevalence of *ARID1A**, JAK* and *KMT2* gene mutations in addition to the common mutations identified in conventional PDAC, but they often have wild-type *KRAS* and *TP53,* suggesting that different drivers are involved in the tumorigenesis of MSI/dMMR PDACs [98,99,100]. It has also been shown that MSI/dMMR PDACs may have significant intra-tumor heterogeneity and may lead to the development of metastatic MSS PDACs and recurrent beta-2-microglobulin (B2M) gene inactivation, which may be associated with tumor resistance to immune checkpoint inhibitor therapy [100].

Although the above-mentioned predictive markers are not common in PDAC patients, the detection of these markers in PDAC patients may have a major impact on the selection of appropriate targeted therapy or immunotherapy and their clinical outcomes. Therefore, the current NCCN Guidelines recommend routine germline testing for all patients with PDAC and the molecular analysis of tumor samples in patients with metastatic PDAC. The American Society of Clinical Oncology (ASCO) guidelines recommend the following treatment options for patients with metastatic PDAC after first-line therapy: (1) In patients with tumors harboring NTRK fusions, treatment with larotrectinib or entrectinib is recommended; (2) pembrolizumab is recommended as the second-line therapy for patients with MSI/dMMR PDAC; (3) in patients who have a germline BRCA1 or BRCA2 mutation and who have received first-line platinum-based chemotherapy without disease progression for at least 16 weeks, options for continued treatment include chemotherapy or PARP inhibitor olaparib [101].

## 7. The Tumor Microenvironment of PDAC

PDAC is characterized histologically by extensive desmoplastic stroma, a hypoxic and immunosuppressive tumor microenvironment (TME) consisting of cancer-associated fibroblasts (CAFs), stellate cells (SC), an extracellular matrix, endothelial cells, myeloid-derived suppressor cells, and low number of tumor-infiltrating lymphocytes (TILs). The heterogeneity of SCs, the molecular mechanisms of SC biology, and therapeutic strategies targeting SCs/stroma in PDACs have been extensively covered in several recent reviews [102,103,104,105]. Once activated, SCs and CAFs can produce various soluble factors such as transforming growth factor β (TGF-β), interleukins, fibroblast growth factor (FGF), stromal cell-derived factor-1 (SDF-1), hepatocyte growth factor (HGF), and galectin-1. Through these soluble factors and cell–cell interactions, SCs regulate the pathogenesis and invasiveness of PDACs [102,103,104]. In addition, SCs also provide the much needed nutrients and metabolites for the hypoxic and nutrient-depleted TME to fuel the energy metabolism of PDAC via autophagy, secreted exosomes, and oxidative stress [105]. The dynamic interplays among the tumor cells, SCs, and other stromal cells are essential in not only tumor growth, angiogenesis, progression and metastasis but also tumor resistance to chemotherapy. Therefore, targeting SCs is a promising strategy and an active area of ongoing research.

Many studies have examined the mechanisms of tumor cell evasion from the tumor-specific immune response in PDACs and their resistance to immune therapy. Several immune subtypes of PDAC have been reported and provide insights into the complex immune landscape of PDAC. For instance, Knudsen et al. identified four immune subtypes (hot, cold, mutationally cold, and mutationally active) of PDAC according to the burden of tumor-specific antigens (neoantigens), as well as immunological and stromal features [106]. They demonstrated that the expression of immunologic markers including PD-L1 (CD274), FOXP3, CTLA4, CD8, CD68, and CD163 were correlated with each other and associated with patient survival [106]. Later, Danilova et al. defined four immune subtypes according to the expression of CD8 and PD-L1: PD-L1+/CD8 high, PD-L1+/CD8 low, PD-L1-/CD8 high, and PD-L1-/CD8 low [107]. In these studies, each subtype has different features such as TILs, TMB, immunologic cell death modulators, stromal fraction, and TGF-β response. Based on the immune cell populations and the mechanisms for evading the anti-tumor immune response, Karamitopoulou classified PDACs into three immunologic subtypes: the immune-escape subtype, which has high FOXP3+Treg cells and M2-macrophages but low cytotoxic T-cells and M1 macrophages; the immune-rich subtype, which has high TILs and tertiary lymphoid structures but a low infiltration of FOXP3+T regulatory cells and M2 macrophages; and the immune exhausted subtype, which has MSI/dMMR, high TILs, and the overexpression of immune checkpoints [108]. These immunologic subtypes correlate with the molecular subtypes of PDAC in that the classical subtype has an immune-rich phenotype while the basal-like (ASC/SCC) subtype is associated with an immune escape phenotype [108,109].

Immunotherapy has limited efficacy for PDAC patients, except those with MSI/dMMR tumors. Although the mechanism underlying the refractoriness of PDAC to immunotherapy remains unclear, emerging evidence suggested that this is likely due to the low TMB and the immunosuppressive TME of PDAC. Compared with the immunogenic NSCLC and melanoma, which have good responses to immunotherapies, PDACs have lower TMB and a lower number of TILs, a lower expression of PD-L1 and PD-1, and higher number of VISTA+ cells [110]. Studies have shown that the presence of a high number of TILs, cytotoxic T cells, and a diverse T cell receptor repertoire are associated with longer survival in PDAC patients while the presence of high CD3+Foxp3+ T cells is associated with shorter survival rates [111,112,113]. Many ongoing studies in both preclinical models and clinical trials are exploring new immune targets and strategies to overcome PDAC resistance to immunotherapy. Their impact on the clinical outcome of PDAC patients remains to be determined.

## 8. Future Perspectives and Summary

PDACs are a heterogenous group of malignant epithelial neoplasms with various histomorphological patterns and complex genetic/molecular profiles. PDACs arise from several distinctive types of precursor lesions, including PanIN, IPMN, IPON, ITPN and MCN. The newly proposed molecular classifications of PDAC based on extensive genomic, transcriptomic, proteomic and epigenetic data have provided significant insights into the molecular heterogeneity and aggressive biology of this deadly disease. Future studies in the following areas may help to improve treatment response and patient survival: 1. High-resolution genomic analysis, such as the single-nucleus RNA sequencing of PDAC samples, may provide more comprehensive roadmaps of tumor heterogeneity for both tumor cells and stromal cells, which may help researchers better understand the dynamic interplays between the tumor cells and TME, the aggressive biology and tumor resistance, as well as develop more effective therapeutic strategies targeting specific molecular/genetic alterations, cellular phenotypes, or multicellular interactions; 2. the integration of histopathology with molecular profiles to provide the better classification of PDACs that will be useful in guiding the selection of optimal treatment regimens; 3. it is unfortunate that only a limited number of predictive markers are currently available for clinical decision makers to select the best possible treatment. Future biomarker-driven clinical trials are critical to the development of new predictive markers for PDAC patients.

## Figures and Tables

**Figure 1 cells-11-03068-f001:**
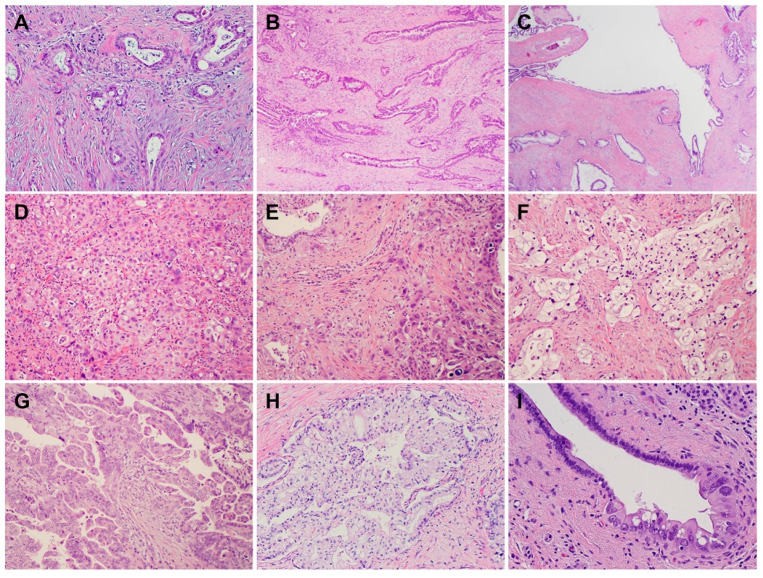
Pancreatic ductal adenocarcinomas (PDACs) with various histomorphological patterns (hematoxylin and eosin stain). (**A**) Moderately differentiated PDAC with extensive desmoplastic stroma; (**B**) moderately differentiated PDAC with interconnecting complex glands embedded in desmoplastic stroma; (**C**) large duct variant of PDAC; (**D**) poorly differentiated PDAC; (**E**) poorly differentiated PDAC intermixed with moderately differentiated areas; (**F**) clear cell variant of PDAC; (**G**) moderately differentiated PDAC showing extensive, complex intra-luminal micropapillary formation; (**H**) cribriform histology with foamy cells; (**I**) pagetoid involvement of pancreatic duct by PDAC (intra-ductal carcinoma).

**Figure 2 cells-11-03068-f002:**
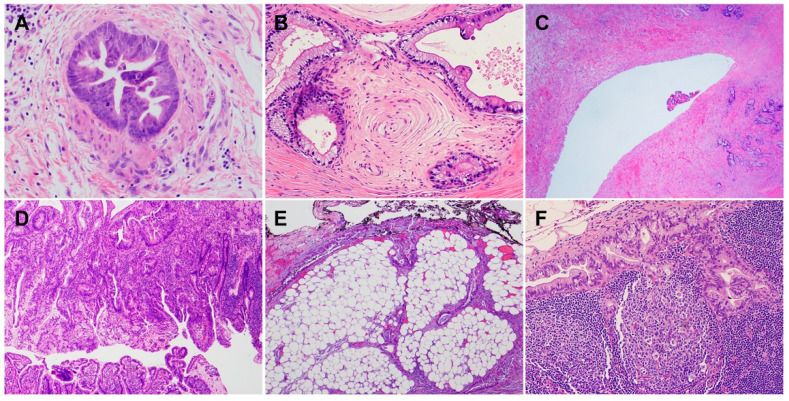
The common histological features associated with aggressive clinical outcomes for pancreatic cancer patients (hematoxylin and eosin stain). (**A**) Tumor invasion into muscular vessels; (**B**) perineural invasion; (**C**) tumor invasion into the wall of superior mesenteric vein; (**D**) PDAC invasion through the muscularis propria of the duodenum and involves the mucosal surface; (**E**) PDAC invades into the peripancreatic and retroperitoneal soft tissue and involves the uncinate margin (marked by black ink); (**F**) metastatic PDAC in a regional lymph node.

**Figure 3 cells-11-03068-f003:**
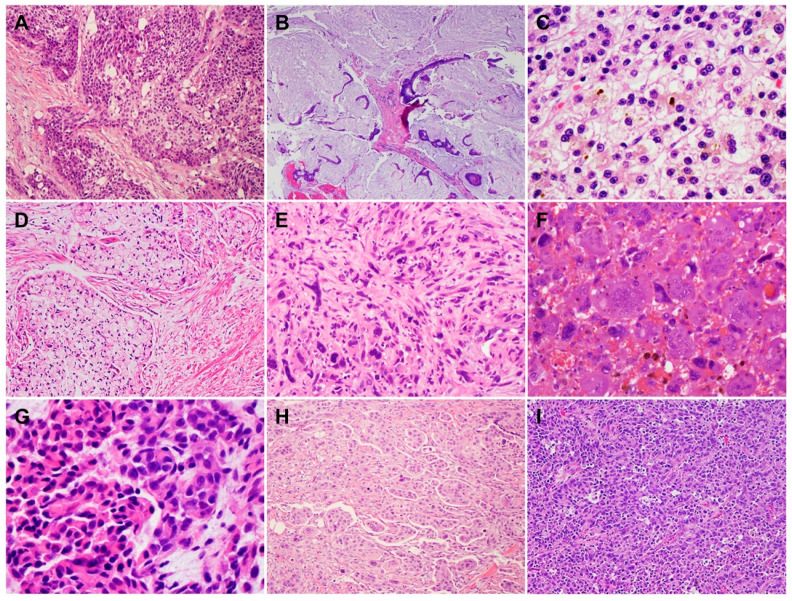
Histological subtypes of pancreatic ductal adenocarcinoma (hematoxylin and eosin stain): (**A**) Adenosquamous carcinoma; (**B**) colloid carcinoma; (**C**) hepatoid carcinoma with bile lakes; (**D**) signet-ring cell carcinoma; (**E**) undifferentiated carcinoma; (**F**) undifferentiated carcinoma with osteoclast-like giant cells; (**G**) undifferentiated carcinoma with rhabdoid cells (rhabdoid carcinoma); (**H**) micropapillary carcinoma; (**I**) medullary carcinoma.

**Figure 4 cells-11-03068-f004:**
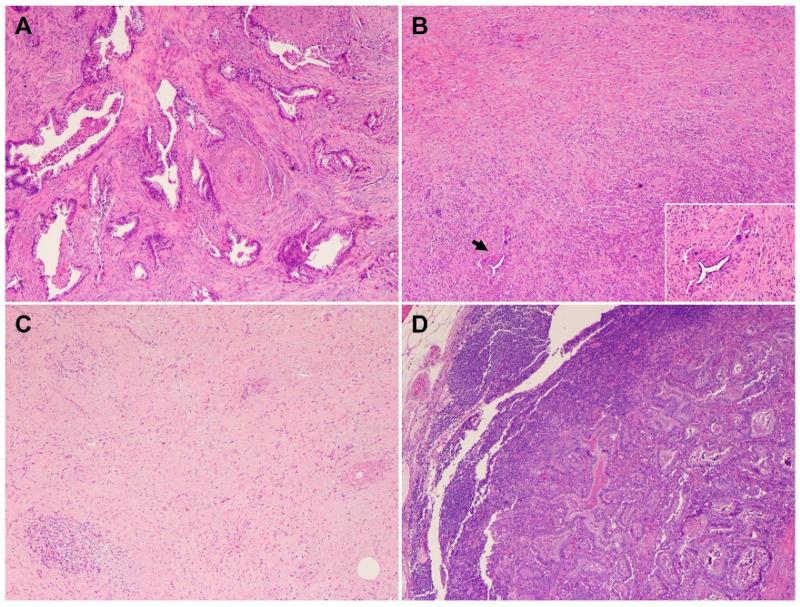
Heterogeneous response of PDAC to neoadjuvant therapy (hematoxylin and eosin stain). (**A**,**B**) Representative micrographs showing a PDAC with an area of minimal response (**A**) and near complete response in other areas (**B**); (**C**,**D**) representative micrographs showing a PDAC with complete response in primary tumor (**C**) but minimal response in the metastatic PDAC in the regional lymph node (**D**).

**Figure 5 cells-11-03068-f005:**
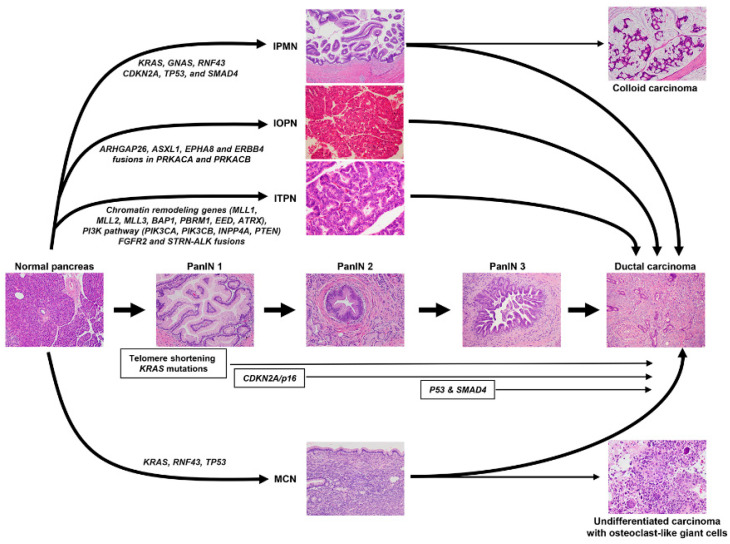
The precursor lesions of pancreatic ductal adenocarcinoma and the common molecular alterations. Abbreviations: PanIN, pancreatic intraepithelial neoplasia; IPMN, intraductal papillary mucinous neoplasm; IOPN, intraductal oncocytic papillary neoplasm; ITPN, intraductal tubulopapillary neoplasm; MCN, mucinous cystic neoplasm.

**Table 1 cells-11-03068-t001:** The subtypes of pancreatic ductal adenocarcinoma and their associated molecular alterations.

Subtypes	Frequencies	Diagnostic Criteria	Specific IHC Markers	Type-Specific Genetic Alterations
ASC/SCC	1–4%	≥30% of SCC	CK5/6, P63, and P40	*UPF1*, *KMT2C*, *KMT2D*, *SMARCA4 (BRG1)*, *KDM6*, and *KDM3*
Colloid carcinoma (CC)		≥80% of CC	CK20, CDX2, and MUC2	*GNAS, ATM,* and *MSI/defective MMR*
Medullary carcinoma	<1%	NA	NA	MSI/defective MMR, POLE
Hepatoid carcinoma (HC)	<1%	≥50% of HC	HepPar-1, Glypican 3, Arginase, and Albumin by FISH	*BAP*1 and *Notch1*
Micropapillary carcinoma (MPC)	<5%	≥50% of MPC	NA	*KRAS*, *TP53*, and *SMAD4*
Signet ring cell carcinoma (SRC)	<1%	≥80% of SRC	NA	*PI3K* and *MEK*
Undifferentiated carcinoma (UC)	1–7%	NA	NA	*CDH1*
UC with osteoclast-like giant cells		NA	NA	*SERPINA3*, *MAGEB4*, *GLI3, MEGF8*, *TTN*, and *BRCA2*
UC with rhabdoid cells	<1%	≥50%	Loss nuclear expression of SMARCB1 (INI1)	*SMARCB1*

Abbreviations: ASC/SCC, adenosquamous carcinoma/squamous cell carcinoma.

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
