# Peer review of "Pancreatic Ductal Adenocarcinoma: Molecular Pathology and Predictive Biomarkers"

_cells, 2022, doi:10.3390/cells11193068_

Round 1

Reviewer 1 Report

The authors have provided us a comprehensive and updated overview of molecular pathology and predictive biomarkers of pancreatic ductal adenocarcinoma. The manuscript is very well written and organized. 

Here are a few minor questions:

1. In figure 2E, it is difficult to appreciate the cancer cells at the inked surface. A picture at higher magnification is recommended.

2. Please provide the full names of the precursor lesions of pancreatic ductal adenocarcinoma before using the abbreviations.

3. The authors may need to briefly discuss the concept of "simple mucinous cyst" and its molecular findings, since there is a grey zone about the mucinous cystic lesion between 0.5 to 1 cm (PanIN lesion <0.5 cm and IPMN lesion >=1cm, as mentioned in the manuscript). 

Author Response

The authors have provided us a comprehensive and updated overview of molecular pathology and predictive biomarkers of pancreatic ductal adenocarcinoma. The manuscript is very well written and organized. 

Here are a few minor questions:

  1. In figure 2E, it is difficult to appreciate the cancer cells at the inked surface. A picture at higher magnification is recommended.

Response: We have replaced the photo for Figure 2E to show the tumor cells at the inked margin.

  1. Please provide the full names of the precursor lesions of pancreatic ductal adenocarcinoma before using the abbreviations.

Response: Corrected. Page 8 line 269 “pancreatic intraepithelial neoplasia (PanIN)” replaced “PanIN”.

  1. The authors may need to briefly discuss the concept of "simple mucinous cyst" and its molecular findings, since there is a grey zone about the mucinous cystic lesion between 0.5 to 1 cm (PanIN lesion <0.5 cm and IPMN lesion >=1cm, as mentioned in the manuscript). 

Response: We thank the reviewer for this important comment. Two paragraphs were added to describe the lesions between 0.5 and 1.0 cm and also the simple mucinous cysts on page 9.

Reviewer 2 Report

Overall, this review summarized histologic and molecular heterogeneity and subtypes of pancreatic ductal adenocarcinoma (PDAC) and its precursor lesions, the immunosuppressive TME and the currently available predictive molecular markers for PDAC patients. This review provides a useful collection for the specialists in the field and may shed the light on finding effective personalized therapies. I would like to recommend its publication after the following of my comments are appropriately addressed.

1. A excellent review should also be critical, summarize important research under the topic, identify consistencies and more importantly inconsistencies, and provide open questions for future research. However, I can’t find a perspective part in this manuscript. For example, in the authors’ opinion, how should the field further explore the diagnostic/prognostic/therapeutic for PDAC through novel single-cell sequencing methods?

2. The authors should perform a thorough formatted and grammatical review for the manuscript and correct all typos. Eg, different font in Introduction part.

3. I’m not sure why the authors didn’t show histologic figures for PDAC histologic subtypes, including medullary carcinoma, micropapillary carcinoma, and undifferentiated carcinoma with rhabdoid cells (rhabdoid carcinoma).

Author Response

Overall, this review summarized histologic and molecular heterogeneity and subtypes of pancreatic ductal adenocarcinoma (PDAC) and its precursor lesions, the immunosuppressive TME and the currently available predictive molecular markers for PDAC patients. This review provides a useful collection for the specialists in the field and may shed the light on finding effective personalized therapies. I would like to recommend its publication after the following of my comments are appropriately addressed.

  1. A excellent review should also be critical, summarize important research under the topic, identify consistencies and more importantly inconsistencies, and provide open questions for future research. However, I can’t find a perspective part in this manuscript. For example, in the authors’ opinion, how should the field further explore the diagnostic/prognostic/therapeutic for PDAC through novel single-cell sequencing methods?

Response: We thank the reviewer for this important comment. Future perspective has added in “FUTURE PERSPECTIVE AND SUMMARY” (Page 13)

  1. The authors should perform a thorough formatted and grammatical review for the manuscript and correct all typos. Eg, different font in Introduction part.

Response: We have carefully checked the format and grammatical errors.

  1. I’m not sure why the authors didn’t show histologic figures for PDAC histologic subtypes, including medullary carcinoma, micropapillary carcinoma, and undifferentiated carcinoma with rhabdoid cells (rhabdoid carcinoma).

Response: Representative micrographs for medullary carcinoma, micropapillary carcinoma, and undifferentiated carcinoma with rhabdoid cells (rhabdoid carcinoma) have been added in Figure 3

Reviewer 3 Report

The review article by Taherian et al on the molecular pathology and predictive biomarkers in pancreatic ductal adenocarcinoma is timely as pancreatic cancer is increasingly seen as one of the most serious problems in cancer therapy. This review article, as the title clearly states, deals with two separate, but of course connected, issues: the pathophysiology of the disease and potential biomarkers. In my view the biomarker issue is handled well; it is up-to-date and comprehensive. However, the pathophysiology (molecular pathology) is incomplete. There is a good section on Genetic Alterations and Molecular Subtypes of PDAC, but the section on the Tumor Microenvironment of PDAC is really rather minimalistic and, strangely and unhelpfully, mixed up with the final summary. An understanding of the nature of the microenvironment, and how it is produced, is essential to a proper understanding of the molecular pathology. This section therefore needs to be revised. It is, for example, simply astounding that the authors have written a section on the micro-environment in which one of the key cells responsible for its production, namely the stellate cells, are never even mentioned! Indeed, as far as I have been able to check, there is no mention of these crucially important cells anywhere in the review and there are no appropriate references to papers dealing with the role of the stellate cells in the generation of the microenvironment. Given, that there has been remarkable recent progress in understanding the properties of these cells (for a recent review see: Physiol Rev 101, 1691-1744, 2021; sections 5,9 and 10), this part of the review (The Tumor Microenvironment of PDAC) needs revision and expansion.

Author Response

The review article by Taherian et al on the molecular pathology and predictive biomarkers in pancreatic ductal adenocarcinoma is timely as pancreatic cancer is increasingly seen as one of the most serious problems in cancer therapy. This review article, as the title clearly states, deals with two separate, but of course connected, issues: the pathophysiology of the disease and potential biomarkers. In my view the biomarker issue is handled well; it is up-to-date and comprehensive. However, the pathophysiology (molecular pathology) is incomplete. There is a good section on Genetic Alterations and Molecular Subtypes of PDAC, but the section on the Tumor Microenvironment of PDAC is really rather minimalistic and, strangely and unhelpfully, mixed up with the final summary. An understanding of the nature of the microenvironment, and how it is produced, is essential to a proper understanding of the molecular pathology. This section therefore needs to be revised. It is, for example, simply astounding that the authors have written a section on the micro-environment in which one of the key cells responsible for its production, namely the stellate cells, are never even mentioned! Indeed, as far as I have been able to check, there is no mention of these crucially important cells anywhere in the review and there are no appropriate references to papers dealing with the role of the stellate cells in the generation of the microenvironment. Given, that there has been remarkable recent progress in understanding the properties of these cells (for a recent review see: Physiol Rev 101, 1691-1744, 2021; sections 5,9 and 10), this part of the review (The Tumor Microenvironment of PDAC) needs revision and expansion.

Response: We fully agreed with the reviewer that an understanding of the nature of the microenvironment, and how it is produced, is essential to a proper understanding of the molecular pathology. Since the functions and biology of stellate cells have been extensively covered by several recent reviews, we have added a brief summary of the role of stellate cells in tumor progression and resistance in the section “The Tumor Microenvironment of PDAC” and four additional references including the reference suggested by the reviewer (references 99-102). We hope that this is OK with the reviewer.

Round 2

Reviewer 3 Report

The revised ms is much improved and is now ready to be accepted